

**PeerJ Hubs**
Published on behalf of

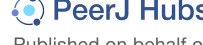
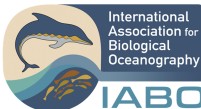
International Association for Biological Oceanography
IABO

# Bio-economic indicators of fisheries: impact of variations in landings and fish size on market prices in Istanbul Fish Market

Taner Yildiz[1], Aylin Ulman[2], F. Saadet Karakulak[1], Uğur Uzer[1] and Nazli Demirel[3]

[1] Faculty of Aquatic Sciences, Istanbul University, Istanbul, Türkiye
[2] Mersea Marine Conservation Consulting, Mugla, Türkiye
[3] Institute of Marine Sciences and Management, Istanbul University, Istanbul, Türkiye

## ABSTRACT

Fisheries are of immense importance to Mediterranean countries, for protein, employment and livelihoods. Studies addressing the factors affecting fish price dynamics are of interest to examine their drivers as prices often dictate target fisheries. This study investigates fish market prices in relation to landings and fish sizes from Türkiye's largest fish market in Istanbul as the study site. A total of 39 wild marine taxa were examined for their landed catch (kg) and average prices (per unit/TRY) from 2006 to 2019. We found fish prices increased from small pelagic to medium pelagic to demersal fish. GAM model results for inter-species tests showed a strong positive effect of local maximum length ($p < 0.01$) and a weaker positive effect of trophic level and vulnerability ($p < 0.1$) on fish market price, but that landings amounts have no significant effect as a single predictor. Monthly price variations of bluefish and bonito were completely different than other species dynamics, as the last substantial commercial medium pelagic fish species left; highest monthly prices were related to the highest monthly landings for bonito and for the non-closure period for bluefish. Market prices as economic indicators for fisheries may have the potential to reveal ecosystem variations as well as socioeconomic drivers. Databases including extensive data for key fish sales centers can be used to help understand fishery dynamics from an ecosystem perspective, especially for data-poor regions like Türkiye.

# INTRODUCTION

Self-renewable fish populations, and the income derived from them grow wild in the sea, but our collection of these stocks should be careful not to compromise their future. Capture fisheries are one of the oldest types of industry (*Campling, Havice & Howard, 2012*); however, there are great disparities between the ways fisheries are managed. Most fisheries research is dedicated to the biological properties of commercial species, but the economic aspects driving fisheries have barely been explored. Fish prices guide target fishery decision-making, and these large-scale removals in turn impact ecosystem

Corresponding author
Taner Yildiz, yldztnr@istanbul.edu.tr

processes (*Simonit & Perrings, 2005*). Catch revenue is the main motive driving fisheries, but as renewable resources, management needs to prioritize the long-term sustainability of stocks, over short term gain (*Lleonart et al., 2003*).

Bio-economic factors such as supply, demand and seafood market prices can be indicators for (i) status of fish stocks (*Quetglas et al., 2016*), (ii) changes in economic interest or targeted policies of the sector (*European Commission, 2021*), and (iii) consumer habits regarding their preferences and purchasing behavior (*Almeida et al., 2015*). Prices at the fish market usually reflect a number of factors, including the aggregate volume of product on the market, the species and their sizes, seasonality, the 'quality' of available catch, the availability of supplies from alternative sources, the number of potential buyers and underlying retail demand conditions, and fuel prices (*Pinnegar, Hutton & Placenti, 2006*). Hence, studies on fish market data regarding prices and landings provide unique information on social trends dependent on the interface between fishers, fishmongers and consumers, as well as novel information on fisheries such as catch lengths, weights, percentage of total sales and market value (*Fortibuoni et al., 2017*). In very basic terms, fish prices generally increase with increasing fish size, as larger fish provide more meat and thus higher profit (*Sumaila et al., 2007*; *Sjöberg, 2015*). Some studies reveal that the fish price-fish size relationship promotes overfishing by removing the largest fish first, before their smaller counterparts (*Pinnegar et al., 2002*; *Tsikliras & Polymeros, 2014*), while other studies indicate the alteration of marine food webs by the removal of high trophic level species resulting in highest attainable prices (*Baeta, Costa & Cabral, 2009*).

Fish markets in Türkiye are regulated by the municipalities for their control and management. The marketing of seafood landed in Turkish waters is generally accomplished through brokers (*Üstündağ, 2013*). Seafood sales take place in the auction hall at select hours determined by the administration, with daily inspections carried out by officials. The percentage rates of the various contributions of the distribution channels are shown in Table 1 (*TURKSTAT, 2022*). From this, it can be clearly seen that the distribution percentages in Sea of Marmara, Aegean Sea, and Mediterranean Sea are quite high for the middle man; whereas in the Black Sea, the middleman rates fall below 50%. This is because most anchovy and sprat in the Black Sea (the bulk of catches) are sent to regional fish meal/oil factories, and are thus not sold at markets. Türkiye is a net exporter in foreign trade of fisheries products, especially for farmed species. In the last two decades, aquaculture production and technology has received strong state investment, and has recently even surpassed the total quantities of wild fisheries, much of which is exported. When the export-import data for 2020 are analysed, exports are 112,000 tons higher than its imports (*Çöteli, 2021*).

Istanbul fish market (IFM) is the largest fish market in Türkiye and holds historical importance in providing oldest fish records dating back to the Ottoman period in the late 19th century (*Deveciyan, 2006*; *Yıldırım & Akyol, 2013*). IFM is the central component of the Turkish fish marketing structure and services different hierarchies of buyers around the country such as wholesalers, retail fish markets, and restaurants (*Yılmaz et al., 2014*). Prior to its current location in western part of Istanbul (Gürpinar, Büyükçekmece district) where it is very far from the city center, IFM was very close the Istanbul's historic center (Kumkapi,

**Table 1  Proportional (%) distributions of averaged marketing channels in Türkiye (*TURKSTAT, 2022*).**

| Marketing channels | Sea of Marmara | Aegean Sea | Mediterranean Sea | West Black Sea | East Black Sea |
|---|---|---|---|---|---|
| Fish oil factory | 6.09 | 3.55 | 0.59 | 10.18 | 46.65 |
| Cooperative | 1.12 | 8.95 | 1.40 | 1.20 | 2.29 |
| Middle man | 80.31 | 78.04 | 85.28 | 44.61 | 46.80 |
| Canning factory | 3.23 | 1.06 | 0.36 | 30.98 | 2.11 |
| Consumer | 3.96 | 5.01 | 6.37 | 3.39 | 0.74 |
| Own consumption | 1.33 | 1.27 | 1.36 | 1.07 | 0.43 |
| Fish farming | 0.50 | 0.08 | 0.25 | 0.62 | 0.13 |
| Other | 3.47 | 2.05 | 4.40 | 7.94 | 0.84 |

Eminönü district) between 1985 and 2015. Back then, according to their policies, fish caught around the Istanbul region (including Sea of Marmara, Istanbul Strait and nearby areas of the Black Sea; Fig. S1) were first sent to the IFM for tax determination and tax collection based on market prices (*Uluskan, 2011*). Although not currently mandatory, this rule has been perpetuated by fishers, and is still exercised today. Today, the commercial fish sold in IFM come from four main sectors: the purse-seiners from the Sea of Marmara, western Black Sea and north Aegean Sea; bottom trawlers from the western Black Sea and north Aegean Sea; beam-trawlers from the Sea of Marmara; and a much smaller artisanal sector when total catches are considered, not number of people employed, using trammel net, gillnet, longline and hook and line fisheries from all these seas. The IFM is a perfectly competitive fish market with many associated companies (104 authorized dealers), unrestricted freedom of entry, and high product diversity. Sales are completed through a bidding auction, with a plan to switch to an electronic clock bidding system in the near future. Sales from the IFM have dramatically expanded in the past 60 years (Fig. S2) owing to improved technologies in transportation, freezing, refrigeration and logistics that makes it an iconic national seafood market incorporating wholesale trading and embodying the national fish culture.

A large portion of fish consumption in Türkiye is fresh fish (75%), followed by fishmeal/oil (14,4%), frozen (4%), canned, and salted (2%) (*Ceyhan, 2019*). No studies have yet addressed the factors affecting fish price dynamics in Türkiye, but as prices often dictate target fisheries, it is important to examine their drivers. This study investigates the relationship between fish market prices and fish sizes from the largest fish market in Istanbul as the study site. Several indicators were tested for 39 marine fish species regularly sold at this market from 2006 to 2019. Annual changes in market landings and fish prices (per kg) were also evaluated monthly. Changes in mean length, mean trophic level and vulnerability of landed species were analyzed according to their landings. The last two remaining medium pelagic fish stocks in the region, Atlantic bonito (*Sarda sarda*) and bluefish (*Pomatomus saltatrix*) (*Daskalov et al., 2020*) were more closely investigated to better understand the relationship between fisher behavior and consumer preferences for landings, fish sizes and prices. Finally, we provide a fish calendar compiled from different sources (*Tezel, 1956*; *Pasiner, 2003*; *İSYÖN, 2022*) to show which fish are chosen in which months, normally based on availability, migrations and closures, especially in the Sea of

Marmara. This calendar can be considered as the backbone showing the national fish supply and consumer preferences.

## MATERIALS & METHODS

### Description of the data

Commercial fish are delivered by fishing boats registered to the Istanbul Provincial Directorate of Agriculture, late each night around 2:00 am to the IFM. Initial product control screening is completed by staff to validate each fishers' declaration. Next, the quantities of each product are registered by number, weight (kg), and value (TRY), before incorporation into the auction area, and these data are regularly sent to both the Turkish Statistical Institute (TURKSTAT) and the Provincial Directorate of Agriculture.

Roughly, about 100 taxa are sold in IFM, and about 70 of those regularly. Fish species comprise the majority of sales (about 95%) while the sales of invertebrates are negligible. We extracted data for fresh marine finfish which have regular landings, thus a commercial importance in fisheries, and excluded frozen and farmed fish so that only local taxa from IFM are incorporated. Finally, a total of 39 wild marine taxa were chosen. Their landed catch (kg) and average prices (per unit/TRY) from 2006 to 2019 were collected from the open-access database in online website of the municipality (https://tarim.ibb.istanbul/tr/istatistik.html; *Municipality, 2020*). In addition to annual data, we also collected monthly landings and price data from January 2009 to December 2019. Inflation in Türkiye generally averages around 10% a year, but since 2018, this rate has increased to about 20%, so changes to prices in the short-term are as affected by this as prices in the long-term. Following a recommendation by *Pincinato & Gasalla (2010)* using an index instead of nominal prices considering inflation rates for analyzing long-term market data for unstable economies, we used a consumer price index to ensure ''real prices'' (inflation-adjusted values). Hence, all fish prices in this study have been realized using the annual and monthly food consumer price index (base year =2003) provided by TURKSTAT.

### Measure of tendencies

Market data for 39 available fish were classified into three categories based on their life and habitat characteristics as five small pelagic fish (anchovy, sardine, *etc.*), nine pelagic fish (bluefish, bonito, horse mackerel, swordfish, tuna *etc.*), and 25 demersal fish taxa (whiting, hake, turbot, *etc.*). Comparing the fish prices by their groupings and fish prices between IFM and nation-wide was performed by using mean annual market price of fish for the period 2006–2019. To graphical illustrate the monthly and annual landings and market prices, we used the means from the monthly, annual landings, and prices. The gross value of production was calculated to show which species provided the most revenue to fishers which helps in understanding their targeting framework, in addition to just price. The gross value of production (GVP) for 12 important species was calculated by multiplying mean monthly catch and mean price.

## Modelling variables effect the price of fish among-species

Generalized additive models (GAMs) were performed to determine which factors correlate to market prices among species. The effect of various predictor variables on the mean market price of fish for the period 2006–2019 (the response variable) were explored by fitting GAM tests. Trophic level, maximum length, common length, vulnerability, landing amount, and local maximum length were used as predictors, as listed in Table S1. Maximum lengths (Lmax), common lengths (Lc: the peak of the population size histogram; that is, the length at which most individuals of the population are sampled), vulnerability indexes (VI) and trophic levels (TL) of all examined species were obtained from FishBase (*Froese & Pauly, 2022*). Vulnerability indexes are provided from a scale of zero to 100, with zero being the lowest vulnerability and 100 being the highest. A literature search was performed to collect local maximum lengths ''local Lmax'' of studied species from the Sea of Marmara, or the nearest adjacent seas if data could not be found for Sea of Marmara as these indices vary regionally. The differences were checked for Lmax from local seas with global values provided from FishBase data, and the Lmax values from local studies are provided in Table S1. The GAM's were generated in R software (*R Core Team, 2019*), using the 'gam' function of the 'mgcv' package (*Wood, 2006*) with Gamma distribution which is used for strictly positive real valued data. Preceding the GAM analysis, predictor variables were logarithmic transformed. The predictor variables were smoothed using the cubic regression spline, then the final model was selected using the Akaike Information Criterion (AIC) value (*Burnham & Anderson, 2002*). In order to determine collinearity among variables a correlation matrix was delivered by using 'corrplot' function of the 'corrplot' package (*Wei & Simko, 2021*). Since several variables were highly correlated (>0.3 correlation coefficients), only some sub-models with two variables were assessed. Although fishing gear type is an important factor in determining the fish prices, this was not incorporated here as the data was not recorded in IFM.

## Price volatility analysis

Monthly price volatility, the degree of variation in prices (*Pincinato, Asche & Oglend, 2010*) describes the standard deviation of logarithmic price returns (*Dahl & Oglend, 2014*). The effects of the conditioning variables on the full sample of price volatilities without any prior groupings was performed to obtain an estimate of the impact of the conditioning variables on price volatility. The coefficient of variation (CV) is the standard deviation divided by the mean, and the negative correlation indicates whether an x% increase in mean landed volume is associated with a <x% increase in the standard deviation (*Pincinato, Asche & Oglend, 2010*). For the regression analysis, three models underwent comparison: (i) accounting for landing volume; (ii) accounting for landings volume and variations in landing volume; and (iii) species groupings by their functional classification (*i.e.,* small pelagic, medium pelagic, and demersal).

## Size effect on prices

Atlantic bonito *Sarda sarda*, and bluefish *Pomatomus saltatrix* are two highly vulnerable medium pelagic iconic species in Türkiye. For both species, market prices vary by length
and are recorded by their size-based names (Table S2). Thus, we also analyzed changes in landings and prices for both small and large size categories for these species. Their monthly variations in landings were related with their migration seasons, and we considered price variations to be mainly related with consumer behavior.

# RESULTS

## Market series analysis

The logarithmic transformation of each species catches to see the natural log of landings highlights the scale of each species' contribution within the multispecies fishery. High percentages of species show that they are primarily targeted where they comprise most of the catch. By comparing percentage weight to the natural log of total landings for all species, anchovy was the dominant landed species in the IFM, followed by Mediterranean horse mackerel, Atlantic bonito, and bluefish (Fig. 1). Anchovy constituted 43% of the landings by quantity in 2019. The mean monthly gross value of production revealed that bluefish and Atlantic bonito had high contributions to fishers, followed by European anchovy and Mediterranean horse mackerel (Fig. 2).

Comparing fish prices by their groupings (Fig. S3), fish prices increase from small pelagic to medium pelagic to demersal fish. Among demersal fish, turbot was most expensive species followed by tub gurnard, European flounder, white grouper, red mullet, and John dory. Bluefish, swordfish, and Atlantic bonito had the highest prices for the medium–large pelagic fish, which all had large-pelagic counterparts landed just a few decades ago. The most inexpensive species were pontic shad, blotched picarel, and European pilchard (Fig. S3).

Landed catch amounts and average market prices varied considerably between 2006 and 2019 (Fig. 3). Changes in average price and landed catch amounts show adverse trends for several fish species, where generally, sequential fish catch declines caused increases in fish prices, noticeable for turbot and anchovy. For most demersal fish, a significant relationship ($p < 0.05$) was found between declining landed catches and increasing sales prices which included John dory, black scorpionfish, sand steenbras, blue whiting, anglerfish, common sole, bogue, European flounder and red mullet, as well as a few medium to large and small pelagic fish including Atlantic horse mackerel, swordfish, and annular seabream (Fig. S4).

For 23 species, the fish prices in 2019 were significantly higher than those in 2006. Many species ($N = 21$) had increasing prices throughout the time-series, ranging from 10% to 255% from 2006 to 2019, with the highest percentage increases found for black scorpionfish and European flounder with 220% and 250%, respectively. Among small pelagic fish, anchovy increased in price 76.4%. Similarly, in medium pelagic fish, little tunny increased 127% in price over the 2006 to 2019 period (Fig. 4). For 11 species, the slope was negative with significant relationships ($p > 0.05$; Table 2).

We compared fish prices between IFM and the nation-wide average for 36 fish species (national price data *via* TURKSTAT, 2022). 13 species had higher market prices in IFM than the Turkish national average such as Atlantic bonito, black scorpionfish, bluefish, brown meagre, European anchovy, European flounder, garfish, John dory, red mullet,

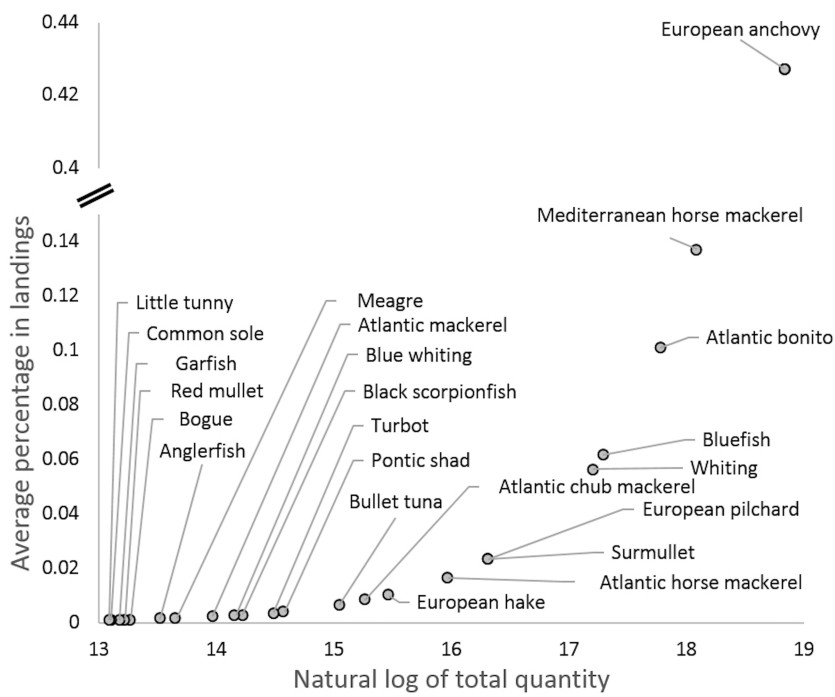

**Figure 1** **Natural log of total quantity landed to average proportion of landed catch.** Only 23 species above 13 natural log of total quantity are presented in the graph, the remaining 16 species below this number were excluded.

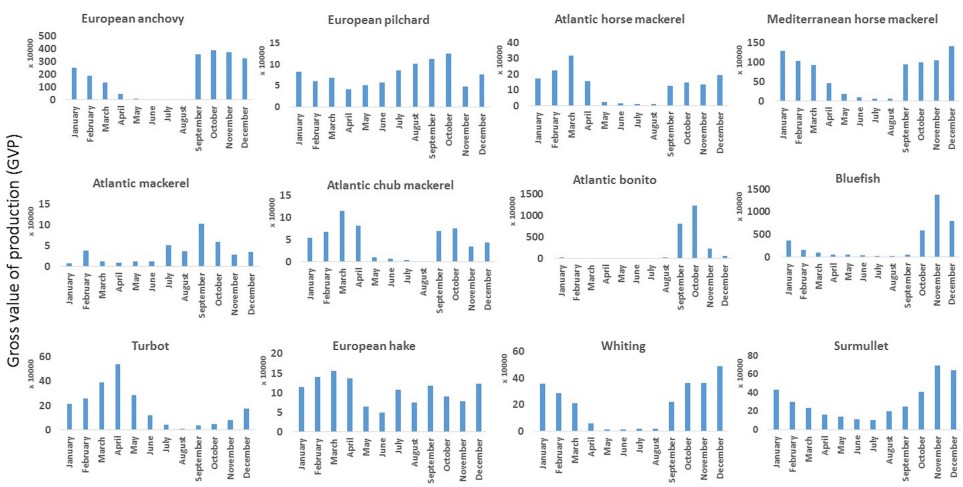

**Figure 2** **Monthly gross value of production averaged between 2009 and 2019 for 12 important species.**

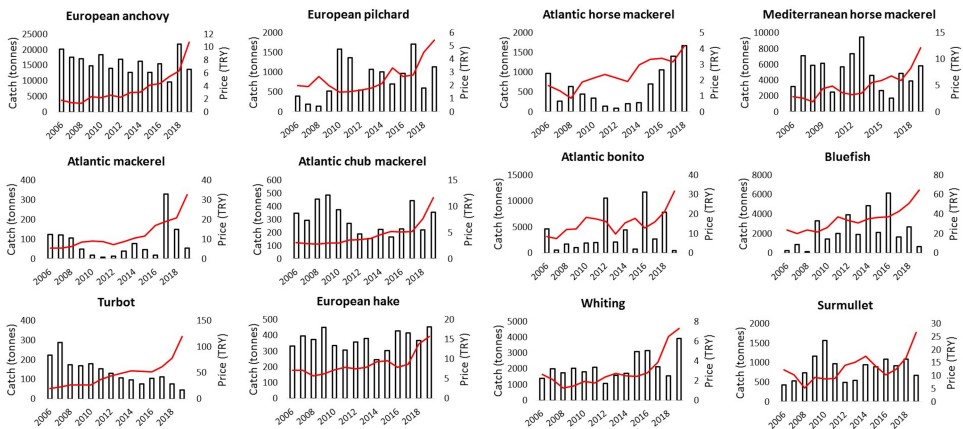

**Figure 3** Annual changes in landed catch (bars) and unit market real prices (red lines) for the main commercial fish species between 2006 and 2019 in Istanbul Fish Market.

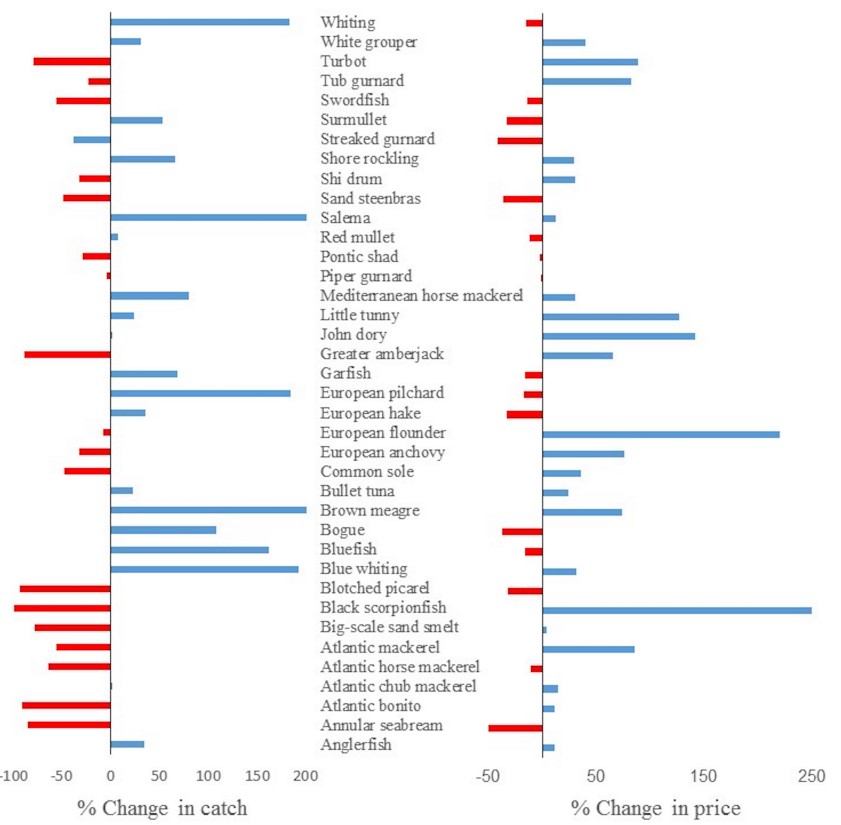

**Figure 4** Percentage changes in average landed catch and real prices from 2006 to 2019.

**Table 2  Relationship between annual landed catch (kg) and average unit real price (in TRY) for each two species assessed here from 2006 and 2019.** Significant $p$-values ($<0.05$) are shown in italics.

| Species | Scientific name | Intercept | Slope | F-statistic | *p*-value | R² |
|---|---|---|---|---|---|---|
| John dory | *Zeus faber* | 22625 | −1015 | 41.83 | *0.0001* | 0.8230 |
| Black scorpionfish | *Scorpaena porcus* | 17331 | −913.3 | 32.61 | *0.0003* | 0.7837 |
| Sand steenbras | *Lithognathus mormyrus* | 26913 | −6493 | 19.09 | *0.0018* | 0.6796 |
| Blue whiting | *Micromesistius poutassou* | 426659 | −108879 | 16.02 | *0.0031* | 0.6404 |
| Anglerfish | *Lophius budegassa* | 98937 | −12120 | 14.88 | *0.0039* | 0.6231 |
| Atlantic horse mackerel | *Trachurus trachurus* | 3069000 | −2070000 | 10.51 | *0.0119* | 0.5677 |
| Common sole | *Solea vulgaris* | 68606 | −3538 | 8.522 | *0.0171* | 0.4864 |
| Swordfish | *Xiphias gladius* | −18489 | 3846 | 5.423 | *0.0448* | 0.3760 |
| Bogue | *Boops boops* | 156633 | −126773 | 5.42 | *0.0449* | 0.3759 |
| European flounder | *Pleuronectes flesus* | 4115 | −165.8 | 5.403 | *0.0452* | 0.3751 |
| Annular seabream | *Diplodus annularis* | 5060 | −2791 | 5.309 | *0.0467* | 0.3710 |
| Red mullet | *Mullus barbatus* | 51826 | −1005 | 5.21 | *0.0484* | 0.3666 |
| European pilchard | *Sardina pilchardus* | 2547000 | −1529000 | 3.806 | 0.0829 | 0.2972 |
| European hake | *Merluccius merluccius* | 523664 | −41574 | 3.565 | 0.0916 | 0.2837 |
| Little tunny | *Euthynnus alletteratus* | 127913 | −40302 | 3.101 | 0.1163 | 0.2794 |
| Meagre | *Argyrosomus regius* | 292390 | −38687 | 3.137 | 0.1103 | 0.2585 |
| European anchovy | *Engraulis encrasicolus* | 26380000 | −7369000 | 2.371 | 0.1580 | 0.2085 |
| Streaked gurnard | *Chelidonichthys lastoviza* | 4878 | −854.9 | 1.967 | 0.1943 | 0.1794 |
| Garfish | *Belone belone* | 62818 | −6023 | 1.903 | 0.2011 | 0.1745 |
| Whiting | *Merlangius merlangus* | 4258000 | −1779000 | 1.751 | 0.2184 | 0.1628 |
| Surmullet | *Mullus surmuletus* | 1332000 | −68748 | 1.692 | 0.2257 | 0.1582 |
| Pontic shad | *Alosa immaculata* | 376836 | −270354 | 1.645 | 0.2317 | 0.1545 |
| White grouper | *Epinephelus aeneus* | −6220 | 1971 | 1.424 | 0.2633 | 0.1366 |
| Medit. horse mackerel | *Trachurus mediterraneus* | 8653000 | −1643000 | 1.368 | 0.2723 | 0.1319 |
| Big-scale sand smelt | *Atherina boyeri* | 31540 | −8085 | 1.265 | 0.2898 | 0.1232 |
| Shore rockling | *Gaidropsarus mediterraneus* | −10047 | 7333 | 1.081 | 0.3256 | 0.1072 |
| Atlantic chub mackerel | *Scomber colias* | 528230 | −129387 | 0.6432 | 0.4432 | 0.0667 |
| Shi drum | *Umbrina cirrosa* | 9493 | −616.5 | 0.5688 | 0.4700 | 0.0594 |
| Atlantic bonito | *Sarda sarda* | 6571000 | −343300 | 0.4919 | 0.5008 | 0.0518 |
| Greater amberjack | *Seriola dumerili* | 7220 | 1148 | 0.3251 | 0.5864 | 0.0444 |
| Bullet tuna | *Auxis rochei* | 430068 | −146031 | 0.3329 | 0.5781 | 0.0356 |
| Blotched picarel | *Spicara maena* | 3564 | 7382 | 0.303 | 0.5954 | 0.0326 |
| Turbot | *Scopthalmus maximus* | 155803 | −2044 | 0.294 | 0.6008 | 0.0316 |
| Tub gurnard | *Chelidonichthys lucerna* | 6481 | −49.79 | 0.2631 | 0.6203 | 0.0284 |
| Piper gurnard | *Trigla lyra* | 1588 | 80.48 | 0.09821 | 0.7611 | 0.0108 |
| Brown meagre | *Sciaena umbra* | 1714 | −54.38 | 0.03717 | 0.8514 | 0.0041 |
| Atlantic mackerel | *Scomber scombrus* | 108377 | −6886 | 0.02295 | 0.8829 | 0.0025 |
| Bluefish | *Pomatomus saltatrix* | 2962000 | −10321 | 0.00446 | 0.9482 | 0.0005 |
| Salema | *Sarpa salpa* | 4825 | −87.27 | 0.00027 | 0.9873 | 0.00002 |

Shi drum, surmullet, swordfish, and tub gurnard. For the remaining species, the prices between IFM and Turkish average were comparable (Fig. S4).

## Species indicators and fish prices

There is considerable variations in the commercial species sales prices assessed here. Mean fish prices (per kg) ranged from the lowest values of 0.89 TRY add price for *Alosa fallax* to the highest values of 19.56 TRY for *Scopthalmus maximus*, while Lmax ranged between 15.2 cm for *Engraulis encrasicolus* to 161 cm for *Xiphias gladius*, and trophic levels ranged from 2.0 for *Sarpa salpa* to 4.5 for *Zeus faber*, *Pomatomus saltatrix*, and *Sarda sarda*.

The GAM model results showed a strong positive effect of local Lmax ($p < 0.01$), and a weaker positive effect of trophic level and vulnerability ($p < 0.1$) on fish market price, while landings amounts were not significant on fish market prices ($p > 0.05$; Table 3, Fig. 5) as single predictors. The examination of predictor variables revealed high positive correlations among Lmax, trophic level, and vulnerability (>0.3 correlation coefficients (Fig. S5). Hence, some models with more than one variable were evaluated by eliminating the correlation between predictors, according to the explained deviance and AIC results having very good performances. Model #6 (M06) best explained the predictors on price, including the effect of local Lmax (Table 3). The effect of landing amount only appeared weak in Model #9 along with trophic level. The overall performance of this model was adequate with a $R^2 = 0.182$, and the resulting model explained 28.8% of the total variance demonstrating that market price was most influenced by maximum fish lengths.

## Bonito and bluefish as the key species

Both bonito and bluefish have higher price variations in their larger rather than smaller fish sizes (Fig. 6), but their prices are also affected by seasonality. Landings of bluefish and bonito peaked in some years, but only for smaller sizes, which reflects their known cycles, whereas larger sized fish landings rarely peaked during the study period. Prices were lowest in peak periods and highest especially for larger fish during the years with low landings. However, monthly variations of bluefish and bonito prices were completely different than all other species examined here. Highest monthly prices were during the highest monthly landings for bonito and during the migration period for bluefish (Fig. 6).

## Monthly variations and volatility

The seasonal industrial fisheries closure in Turkish waters occurs between 15 April and 31 August annually. During this 135-day closure period, monthly averaged pelagic fish landings decreased, as expected. From 2006 to 2019, the decrease in average landings during the industrial fishing closure period was approximately 90% for anchovy, horse mackerel, Atlantic mackerel, Atlantic chub mackerel, bluefish, and bonito (Fig. 7). Interestingly, monthly averaged unit prices slightly increased for small and medium pelagic, and clearly increased for demersal fish landed during the closure period (Fig. 7, Fig. S6).

Log mean volume landed and log coefficient of variation were negatively correlated, which demonstrates that higher landed volume is associated with a lower coefficient of variation (Fig. 8). Variations in landings were highly significant ($R^2 = 0.30$, $p < 0.001$). The results indicate that a 1% increase in the coefficient variation of landings was associated

**Table 3** Summary of the GAM model used to test the effect of various variables on the fish market prices.

| Model No | Models | DE (%) | GCV | AIC | $R^2$ |
|---|---|---|---|---|---|
| M01 | s(Troph)[+] | 25.9 | 0.905 | 216.34 | 0.062 |
| M02 | s(Lmax)[*] | 19.2 | 0.767 | 212.88 | 0.064 |
| M03 | s(Lc)[*] | 13.6 | 0.814 | 202.66 | 0.066 |
| M04 | s(Vulnerability)[+] | 7.58 | 0.864 | 217.16 | 0.043 |
| M05 | s(Landings) | 3.75 | 0.900 | 217.84 | 0.006 |
| M06 | s(Llocal)[**] | 28.8 | 0.695 | 208.44 | 0.182 |
| M07 | s(Landings)+s(Lc)[*] | 18.3 | 0.817 | 202.68 | 0.067 |
| M08 | s(Landings)+s(Lmax)[*] | 20.9 | 0.782 | 213.54 | 0.011 |
| M09 | s(Landings)*+s(Troph)[**] | 16.3 | 0.826 | 214.45 | 0.072 |
| M10 | s(Landings)+s(Llocal)[**] | 29.4 | 0.725 | 209.94 | 0.165 |
| M11 | s(Landings)+s(Vulnerability) | 9.6 | 0.893 | 218.35 | 0.036 |

Notes.
Troph, trophic level; Lmax, maximum total length in cm; Lc, common length in cm; Llocal, Local Lmax in cm; Landings in tonnes; DE, Deviance Explained; GVC, minimized generalized cross-validation score; $R^2$, the proportion of the variation explained by the variable.

[*]$p < 0.001$.
[+]$p < 0.05$.
[**]$p < 0.1$.

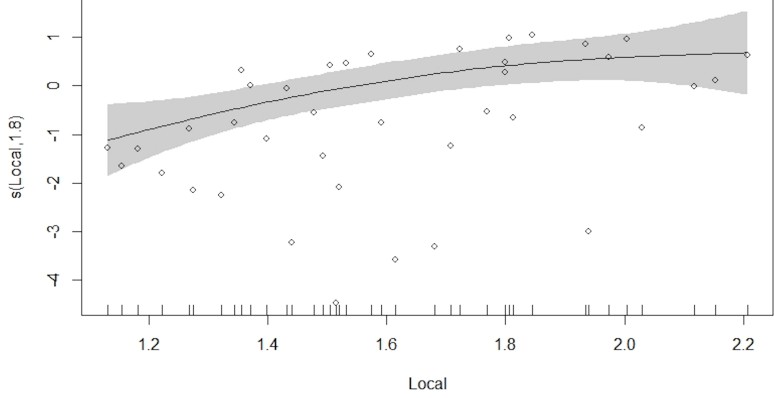

**Figure 5** Coefficients of the GAM (black line) with 95% CI at $p < 0.05$ (grey area) for price against additive terms used in pairwise models.

with a 0.30% increase in volatility. The data were then aggregated for species into habitat groups to indicate how volatility (standard deviations of log-returns) varies across different categories and each species. Demersal species had a lower mean volatility (0.13) in fish prices compared to the small (0.17) and medium pelagic fish (0.18), with pelagic bluefish being the most volatile (0.29).

For the regression analysis, three models were compared: the first without any conditioning variables; the second one accounting for landings volume; and the third accounting for both landings volume and variations in landings volume. The first model, without any conditioning variables found a significant difference in volatility across species

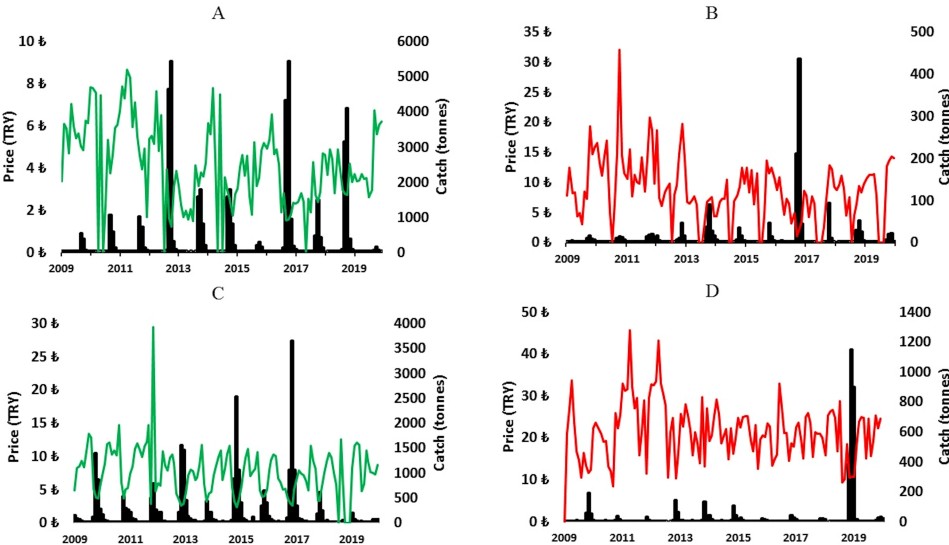

**Figure 6** Small bonito (A), larger bonito (B), small bluefish (C), and larger bluefish (D), showing inter- and intra-annual changes in landed catch (black) and unit real prices (green (A & C) and red (B & D)) from January 2009 to December 2019.

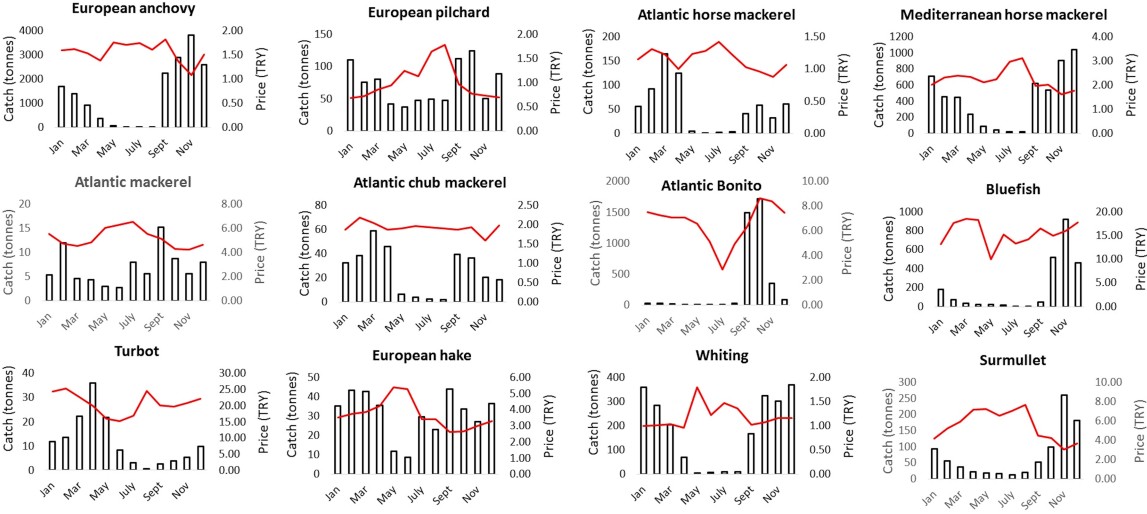

**Figure 7** Average monthly changes in landed catch (bars) and unit real prices (red lines) for the key fish species between 2009 and 2019 in Istanbul Fish Market.

groups (Table 4). Medium pelagic fish have a positive effect on volatility, while demersal fish have a negative effect. For the second and third models, no significant differences in volatility across species groups were found over the 2006 to 2019 study period with conditioning variables. Consistently, medium pelagic fish had the highest volatility, but this was not significant.

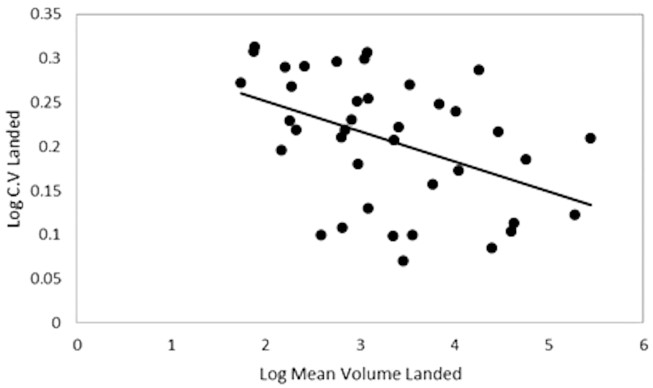

**Figure 8** Log mean volume landed plotted against log coefficient of variation (standard deviation/mean catch) of the commercial species landed catch.

**Table 4** Regression analysis results showing differences in volatility across species groups among the three models.

| | Model 1 | | | Model 2 | | | Model 3 | | |
|---|---|---|---|---|---|---|---|---|---|
| | Coef. | *t*-value | *p*-value | Coef | *t*-value | *p*-value | Coef. | *t*-value | *p*-value |
| Conditioning Variable | | | | | | | | | |
| Log Mean volume landed | | | | 0.014 | 1.569 | 0.125 | 0.031 | 2.972 | 0.005* |
| Log C.V. of volume landed | | | | | | | 0.308 | 2.771 | 0.009* |
| Dummy Variable | | | | | | | | | |
| Small pelagic | 0.009 | 0.422 | 0.676 | −0.019 | −0.767 | 0.448 | −0.0348 | −1.492 | 0.145 |
| Medium pelagic | 0.037 | 2.142 | 0.039* | 0.032 | 1.905 | 0.064 | 0.0202 | 1.234 | 0.225 |
| Demersal | −0.033 | −2.199 | 0.034* | −0.023 | −1.332 | 0.191 | −0.0037 | −0.210 | 0.835 |

## DISCUSSION

In this study we examined indicators and trends affecting the price of fish from 2006–2019. The most expensive fish for sale in IFM were demersal fish, led by turbot, followed by tub gurnard, European flounder, white grouper, red mullet, and John dory, but also only pelagic bluefish with a higher mean price Most demersal fisheries showed a relationship between lower catches and higher prices progressively over the study period, especially for turbot. We also found 31% (12/39 species) had significant and powerful relationships between landed catch amounts and average prices. This price variation is especially more prominent for demersal and medium pelagic fish during the industrial seasonal closure period, likely due to lower availabilities of fish caught predominantly industrially. Maximum length had the strongest effect on market prices, with trophic level and common length having a lower effect. When correlations between predictors were removed, the strongest confidence level was found for local maximum length as the best predictors on unit price, demonstrating that market price is influenced by larger fish.

Other studies have also found size to be the most important attribute in setting the price for many species (*Carroll, Anderson & Martínez-Garmendia, 2001*; *Asche & Guttormsen, 2001*; Kristofferson & Rickertsen, 2004). Although the effect of fish size on fish prices

has long been understood (*Gulland, 1982*), this important relationship has indeed gone overlooked in fisheries science (*Sumaila et al., 2007*), but is a great predictor of profit, and hence targeted species, as fishers try to generate the highest profit available to them. *Tsikliras & Polymeros (2014)* also showed that larger fish related to higher prices, and their selective removals may have contributed to overfishing. However, in our GAM model the explanatory power of these predictors is not very strong. For example, here we have an unexpectedly lower unit price for swordfish, aside from it having the largest Lmax of the species examined in this study. In addition, we found that market conditions do not always cause larger species to be fished more, local demand can still cause smaller species to have high prices.

## Consumer preferences

As *Clark (2006)* pointed out, when fishers caught abundant fish that overloaded their hauls, fish prices generally decreased, and when catches were low, prices increased. These adverse relationships can be used to detect a 'scarcity effect' through signs of over-exploitation (*Clark, 2006*). It is to say that, today, consumer behavior is one of the main drivers behind global resource exploitation (*Richter & Klöckner, 2017*) and is strongly related to cultural habits, income, quality and supply, as well as substitution (*Gobillon, Wolff & Guillotreau, 2017*). The size selectivity through consumer and fisher behavior affect the health of many marine ecosystems and demonstrates the need for a paradigm shift in fish preferences (*Mangel & Levin, 2005*; *Tudela & Short, 2005*). For example, a shift that favours small pelagic fish over large ones, which are in fact much healthier options due to increased Omega-3 oils (*Majluf, de la Puente & Christensen, 2017*). In this study, anchovy had highest market prices in September at the commencement of the industrial fishery season, likely attributed to consumer demand for anchovy rising after the seasonal prohibition. Also very importantly, anchovy is the most primary fish for low-income people as a protein source. Furthermore, differences in average prices across species, like turbot, flounder, bluefish, Atlantic bonito, and grouper are considered as the highest quality fish which are bought by the top tier of restaurants and hotels. Most pelagic fish are caught in specific seasonal periods *i.e.,* during their migrations, and consumers prefer the best optimum time to eat certain types of fish, when they taste best. Our results on fish consumer preferences strongly coincide with the well-known "which month which species calendar" in Türkiye (Fig. 9), so consumers are well educated in this.

This study showed bonito and bluefish to have the highest price volatilities, likely resultant from oscillations in catches every two to five years, which then saturate the market, resulting in lower prices from increased supply. Higher prices were found when their catches were scarce, indicating a scarcity effect relationship on prices. In and around Istanbul, the culturally preferred and iconic species is bluefish, and as its catches decline, its price continues to increase, attributed to its increasing scarcity. Contrary to the trends of other species, the abundant bluefish and bonito catches in autumn causes the prices of these two species to increase rather than decrease. Many people of Istanbul await the arrival of these two species. For the first time in 2021, a consortium consisting of the Istanbul Chamber of Commerce, Istanbul Provincial Directorate of Agriculture and Forestry, Istanbul Trade

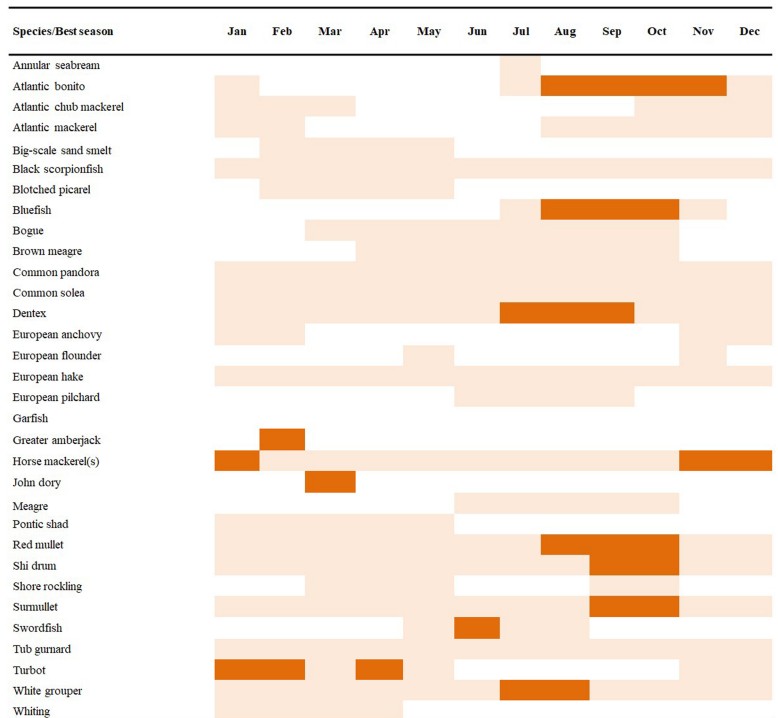

**Figure 9** **Informative calendar for the best month for fish species for consumers in Turkish waters.**

Provincial Directorate, the biggest fisheries cooperative—Sür-Koop, and Turkish Marine Research Foundation—TUDAV applied for a certification of Geographical Indication as a Traditional Specialty for the ''Istanbul Strait Bluefish'', to be one of Istanbul's potential Geographical Indication products (*Ankara Chamber of Commerce, 2021*). Geographical indication is defined as an industrial property right describing a product originated from any region or attributable to any region due to its quality, reputation or other characteristics (*Turkish Patent and Trademark Institute, 2022*).

The increase in global food prices that began in 2003 has been the subject of many scientific analyses (*Headey & Fan, 2008*; *Ivanic & Martin, 2008*; *de Hoyos & Medvedev, 2011*). Less studied, but in some ways more profound, is the increase in fish prices revealed the average annual rate of increase in real fish price, was 1.4% between 1990 and 2002 and a slightly lower 0.9% between 2003 and 2014 (*Nguyen & Kinnucan, 2018*). Food price inflation has increased persistently in recent years in Türkiye with a widening divergence from international food price inflation. Between 2012 and 2019 average food prices declined by 17% in the international markets while they increased by 130% in Turkish markets (*World Bank, 2020*). According to our study, increases in Turkish fish prices were over 200% for some species. The portion of the fuel cost compared to the total costs reached 30% for the Turkish purse seiners (*Koyun, Yıldız & Ulman, 2022*), which capture 90% of total catches. These fuel cost increases really inflated their operational and hence production costs (*Guillen & Maynou, 2016*). Türkiye has one of the lowest annual seafood consumption

rates amongst Mediterranean countries ranging between 5–6 kg per person (5.4 kg in 2016, 5.5 kg in 2017, and 6.1 kg in 2018; 6.2 kg in 2019, 6.7 kg in 2020; TURKSTAT, 2021). Within the last five to ten years, many restaurants are no longer printing fish or seafood prices on menus as there is way too much variation for them to keep up with.

## Data limitations

The data set used in this study had some limitations which require explaining, and were also reflected in the analysis outputs. Firstly, the species-specific information on trophic levels and common lengths were collected from FishBase due to the absence of local studies, which does not properly reflect local differences, especially for trophic levels which are prescribed based on diets. However, maximum length values were wherever possible from local studies and were then cross-checked with FishBase data; when the differences were clear, we used the local values (*e.g.*, the Mediterranean horse mackerel has a much lower Lmax from our region since the Marmara Sea-Black Sea stock is a different stock from the Mediterranean stock; *Demirel & Yüksek, 2013*). Secondly, unit price is a measure of the relative benefits society gets from fish compared to other products (*Morey, 1980*); to properly understand the whole supply–demand chain and the price index better, both sides from the supply and demand sides would have to be thoroughly investigated which is very complex to do in practice. Economic indicators of a fishery itself (prices, catch per unit effort (CPUE), employment, investment, productivity, income distribution) may therefore act as a backdrop to studies on how the fishery responds to price-driven changes specific to the sector, ultimately using those prices to predict changes in the fishery (*Simonit & Perrings, 2005*). However, we only had fish price as an indicator available from the IFM data which indeed was a limitation. Finally, we could not examine the effect of fish quality and size grades from the available data. Thus, to produce a finer-tuned analysis of factors affecting supply and demand, data on more indices would have to be collected.

## Management implications

Under an ecosystem approach, some regulations should be implemented to protect the fishery resources as the fisheries of the Sea of Marmara are declining in all key aspects, specifically in landed amounts (*Demirel, Zengin & Ulman, 2020*), total catch incomes, number of target species (*Ulman et al., 2020*; *Saygu et al., 2023*), and average fish sizes (*Demirel et al., 2023*). To effectively manage fish stocks, control measures need to be applied to limit catches. The other way to rebuild fisheries, by limiting capacity, did not work, there were four national buyback programs in the 2010s, but the capacity of the large-scale fishing fleet still increased despite these measures as mostly small-scale vessels were retired (*Ekmekci & Ünal, 2019*; *Ünal & Güncüoğlu-Bodur, 2020*). Additionally, a biological monitoring program should be established for the commercial fishery resources so that critical habitats for juveniles and new recruits are protected to help the fisheries recover. *Yıldız, Ulman & Demirel (2020)* proposed the establishment of a "Real Time Monitoring" system to dynamically protect critical stocks (such as spawning or nursery areas) which may include a combination of live video feedbacks, onboard observers and the Vessel Monitoring System (VMS).

From a socio-economic perspective, some regulations should be implemented to protect fishers and their wellbeing. In Turkish fisheries, especially from the Sea of Marmara, fish sales are dominated by middle men. Prior to opening of the industrial fishing season, fishers receive cash advances from the middlemen for maintenance, repair costs and for the purchase of new equipment. During the fishing season, in return for their debts, fishers give almost all their caught fish to the middlemen. This type of debt repayment system impairs the fisheries efficiency for better valuation of fish prices, as the middle men are provided with nearly exclusive control for price fixing. In addition, fishery cooperatives are abundant in Türkiye, yet most are lack proper infrastructure for their own marketing capabilities; Thus they would benefit from institutional support to make their sales area operational enabling them to determine their own fish prices.

The role of the fish market is certainly central to fisheries management and is not just a place where prices are decided upon. Fish markets should also function as a consumer education and management area, *i.e.,* by helping consumers make decisions through eco-labelling (*Poindexter, 2015*). In this study, data obtained from IFM proved that the larger fish can be used as a proxy of selective fishing pressure. Thus, consumers should be provided the proper knowledge towards better decision-making for purchasing sustainable seafood stocks. *Rodriguez, Calvo-Dopico & Mourelle (2021)* showed that a 1% increase in stock health translates to 5% reductions in prices, proving that rebuilding and conservation policies are effective tools for ensuring food accessibility. In this case, price-fixing may prevent high price volatility from increasing or decreasing too fast.

Under the current technological innovations that are entering the sector, it is necessary to improve the fish market structure by using electronic systems in marketing and auction processes, and ensuring the traceability of fish products to the end consumer. Such systems can also facilitate electronic data storage for each entire transaction path. The primary purpose of wholesale fish markets is improved efficiency in the food distribution pipeline *Işık (2020)*. Thus, the following important functions are also expected to be provided: (i) physical exchange of products; (ii) classification of products according to standard criteria; (iii) formation of an equilibrium price; (iv) information exchange between buyers and suppliers; and (v) risk management and hedging against price fluctuation risks (*Tollens, 1997*).

## CONCLUSION

In this study, 14-years market time-series analyses were used to determine indicators of fisheries impacts on the variations of market prices. While, 12 key species had annual average price and landings significantly negatively correlated, no positive correlations, on the other hand, between monthly prices and landings were determined. We interperet those results to suggest intra-annual variation is signaling continuous overfishing, while inter-annual variation is likely linked to increasing seafood demand and resource scarcity. The two last remaining medium pelagic fish species, the iconic bluefish and bonito, showed their high price volatilities and highest unit prices during their highest monthly landings. We conclude that the consumer behavior which is strictly related to a regional

and cultural context for demanding any seafood product is one independent driver for price increases. Although Türkiye lacks a developed fishery economics model to follow, decisions should be made based on bio-economic future models combining biological and economic application models along with consumer preferences. Market prices as economic indicators for fisheries have the potential to reveal variations in ecosystem dynamics. Inclusion of this information in databases so that they can be analyzed may help to better understand such dynamics, especially for data-poor regions like Türkiye.

## ACKNOWLEDGEMENTS

The authors would like to thank Dr Hakan Bektaş from the Istanbul University Faculty of Economics.

### Funding

The authors received no funding for this work.

### Competing Interests

Aylin Ulman is employed by Mersea Marine Conservation Consulting.

### Author Contributions

- Taner Yildiz conceived and designed the experiments, performed the experiments, analyzed the data, prepared figures and/or tables, authored or reviewed drafts of the article, and approved the final draft.
- Aylin Ulman conceived and designed the experiments, performed the experiments, authored or reviewed drafts of the article, and approved the final draft.
- F. Saadet Karakulak conceived and designed the experiments, authored or reviewed drafts of the article, and approved the final draft.
- Uğur Uzer performed the experiments, authored or reviewed drafts of the article, and approved the final draft.
- Nazli Demirel conceived and designed the experiments, performed the experiments, analyzed the data, prepared figures and/or tables, authored or reviewed drafts of the article, and approved the final draft.

### Data Availability

The raw data for fishery landings and market prices for 39 species are available in the Supplementary File.

### Supplemental Information

Supplemental information for this article can be found online at http://dx.doi.org/10.7717/peerj.15141#supplemental-information.

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
