# Peer review of "Bio-economic indicators of fisheries: impact of variations in landings and fish size on market prices in Istanbul Fish Market"

_PeerJ, doi:10.7717/peerj.15141_

## Round 0.1 · original submission · Major Revisions

Dear Authors

The reviewers have commented on your manuscript. You can find attached reports. Based on the comments and suggestions of the expert reviewers, unfortunately, a major revision is needed for your article.

I would like to request you check and correct the manuscript step by step based on the reports.

Sincerely yours

Reviewer 1 ·

Basic reporting

Clear and unambiguous, professional English used throughout.
The article is overall well-written and clear. Although, in counted occasions the authors have used complex expressions that hamper the readability of the paper and might induce confusion. For example, in line 184, where it says “However, all relationships between market price and landed catch were not found to be significant”, it would be clearer to say: “some of the 11 relationships are significant”, which would mean the same, but in a more clear and direct way. This is not a major issue, as it happens counted times along the text, but I would recommend the authors to go through the text and eliminate unneeded passive voices and complex structures in favour of a more direct and clear language.
Literature references, sufficient field background/context provided.
The theoretical justification of the paper is sound and enough. Nevertheless, I have observed some places were the authors make a point and use a reference to justify it, but the point of the paper is about something different. The paper might say/suggest that idea in some paragraph, but the general idea of the paper is other. I would suggest the authors caution with their referencing in this aspect, and in this case, not only providing the reference, but a brief explanation on why you think that paper is supporting your point. Examples of this issue are lines 38 and 386.
Professional article structure, figures, tables. Raw data shared.
I am unsure if all the data has been shared.
The data included in the “raw data” xlsx folder consists in annual series for prices and quantities for different species. In the supplementary docx file there are tables with some average values for variables like vulnerability and common length. This is all the data that I have received.
In the manuscript, the authors talk about “intra- and inter-annual market time-series” (316), that “Changes in mean length, mean trophic level and vulnerability of landed species were analysed according to their landings” (84), or about “monthly averaged pelagic fish landings mainly decreased” (212). This suggest that they have performed analysis with data that has not been provided. If there is any issue with confidentiality, please inform so according to journal policies, but specify what data you are using, it is quite unclear, as I will discuss later.
I would also like to suggest simplifying the abstract. The part explaining the results and contributions seem too long (it is half of the abstract) and gives too much information.

Experimental design

Research question well defined, relevant & meaningful. It is stated how research fills an identified knowledge gap.
Although the paper is quite relevant, I think the contribution should be stated in a clearer way. In the abstract, the research gap is described as “To date, no studies have addressed the factors affecting fish price dynamics in Turkey, but as prices often dictate target fisheries, it is of interest to examine their drivers”. First, the dynamic price-size has been studied in the literature in multiple ways, so what is your contribution here, in one phrase? Second, “no studies have addressed the factors affecting fish price dynamics in Turkey”, ok, so why studying it? Maybe the Turkish case can be extrapolated to other Mediterranean regions, it is a major player in X species or market, or simply you have access to a privileged dataset or case study, where you find meaningful insight. Justify the relevance of your sample. And declare what is the knowledge gap you are addressing, and your main hypothesis/objective.
Methods described with sufficient detail & information to replicate.
I do not doubt on the validity of the analysis, but I firmly think that the methodology section must be re-written, specifying clearly what has been done. My main concerns are two:
The data: As I have already stated, the authors only provide annual series for two variables across different countries. But later in the text they mention an analysis performed with series for other variables (like size), monthly series, and even suggest transaction series. Please, clearly specify the data you are using, and provide it (or a statement justifying not doing so). At this point I have no idea which data have been used for each analysis. If I understood it correctly, the authors have put together a quite interesting and complete dataset, so explain it, and I would advise emphasizing its value.
The methods: I understood that the main analysis relies in a GAM model, but after extensive reading of the methods, results, discussion, and tables, I am unable to determine what has been exactly done. I am sorry for the harshness, but I find this part of the manuscript is written in a confusing and incomplete way. Variables that seem to be included in the model are not, but are included in parallel analysis, the data used for each analysis is unclear, I am not able to discern what has been done in each analysis, and I do not see the overall narrative. Please, rewrite this part in a clear and comprehensible way, stating clearly all the steps and justification of each analysis done. At this moment, it is very difficult to understand, and I can not revise it with confidence.
Also, when using real prices, indicate the base year.

Validity of the findings

Conclusions are well stated, linked to original research question & limited to supporting results.
Managerial implications are provided, which is a nice addition. Nevertheless, the said implications are not based on the analysis, but more of a managerial commentary on some of the issues.
I do believe there are interesting findings regarding price-size dynamics and price volatility, like the evidence found of smaller species having higher prices, but the conclusions are centred on other issues, that are not supported by the analysis:
“The economic analysis in this study cannot itself offer a sharp solution, but can be used to show the origin of the problem, namely the faults of an open access regime.” -> this is literally the first time in the manuscript where open access in mentioned. This goes though a different branch of the literature.
“We found significant negative correlations between price and quantity for only 12 key species which signals continuous overfishing” -> do provide some overfishing data in the discussion or somewhere if you want to conclude this. You might find useful indicators in FishSource or RAM Legacy, or in related literature. If not, be careful with your statements. I know there is a relationship between raising prices and overfishing, but to conclude this, you need to prove it, not assume it.
“Market prices as economic indicators for fisheries have the potential to reveal variations in ecosystem dynamics.” -> same as before, I would not conclude this without providing evidence, but it could be stated as a future line of research.
Overall, I would advise rewriting the conclusions, with exactly what the analysis performed is contributing.
Also, I would not recommend introducing new references in the conclusion (line 386).

Additional comments

Overall, I consider that the manuscript is an interesting piece of research. The dataset, if I have understood correctly what has been done, is quite complete and with a lot of potential, and interesting points are made. Although, I consider that the narrative of the paper is diffuse, and the methodology in not well stated, both for reading and for reviewing purposes. Therefore, I must recommend a major revision of this points, after which I would be glad to revise the paper again. I would like to encourage the authors through the review process despite my criticism, I do believe they have made a research with potential, and that the effort will be worth it.

Reviewer 2 ·

Basic reporting

no comment

Experimental design

no comment

Validity of the findings

no comment

Additional comments

Dear Author/s;
Thank you for the chance to read your paper. It is an interesting article that discusses the relationship between fish market prices and fish sizes from Turkey’s largest fish market in Istanbul. Some problems in this manuscript need to be corrected. Please find my other comments in the PDF file as an attachment. Hopefully, they are helpful for your revision. Please review the pdf file carefully and make the necessary corrections.
Please find my other comments below;
1. Supplementary Table S1 is not cited in the text.
2. it should be given a List of variables used for the GAM modeling.
3. Why are some figures given between 2009 and 2019 (Figures S7, 2,6,7) and other figures from 2006 and 2019? It needs to be checked and corrected.
4. Some references are not listed in the bibliography, likewise references in the bibliography are not cited in the text.

Annotated reviews are not available for download in order to protect the identity of reviewers who chose to remain anonymous.

---

## Round 0.2 · accepted · Accept

Dear Dr. Demirel

I am pleased to inform you that your article "Bio-economic indicators of fisheries: Impact of variations in landings and fish size on market prices in Istanbul Fish Market" has been accepted by PeerJ.

Sincerely yours

Reviewer 1 ·

Basic reporting

No comment

Experimental design

No comment

Validity of the findings

No comment

Additional comments

All my previous comments have been addressed, and the paper has improved much. I feel that it is ready for its publication in the journal.
Having said that, I will comment that it seems that the authors misinterpreted a comment of my previous review. When I commented that that sample selection (Turkey) was not justified, I did not mean that the country is not relevant for the case, on the contrary, I know it is. My point is that it is not currently justified on the manuscript. If possible, I would advise the authors to include their answer to the comment (Turkey is a major player in Mediterranean fishery, number 1st…) in the manuscript, as it will show the reader why the country is relevant for the conclusions.
In any case, my congratulation to the authors, and kind regards.